# Follow-Up after Curative Surgical Treatment of Soft-Tissue Sarcoma for Early Detection of Recurrence: Which Patients Have More or Fewer Visits than Advised in Guidelines?

**DOI:** 10.3390/cancers15184617

**Published:** 2023-09-18

**Authors:** Anouk A. Kruiswijk, Laurien S. Kuhrij, Desiree M. J. Dorleijn, Michiel A. J. van de Sande, Leti van Bodegom-Vos, Perla J. Marang-van de Mheen

**Affiliations:** 1Department of Biomedical Data Sciences, Medical Decision Making, Leiden University Medical Center, Albinusdreef 2, 2333 ZA Leiden, The Netherlandsp.j.marang-van_de_mheen@lumc.nl (P.J.M.-v.d.M.); 2Orthopedic Surgery, Leiden University Medical Center, Albinusdreef 2, 2333 ZA Leiden, The Netherlands

**Keywords:** soft-tissue sarcoma, follow-up, surveillance, guidelines, personalized care, overuse

## Abstract

**Simple Summary:**

Soft-tissue sarcoma patients need regular check-ups after surgery to detect disease recurrence as early as possible. However, the current guidelines for follow-up are not based on strong evidence and do not consider individual patient or tumor characteristics. This has led to debates, especially due to concerns about cost, radiation frequency, and possible over-testing. The goal of this study was to see how often patients received the recommended follow-up visits and to identify which type of patients received more or fewer visits than advised. The results show that only 24% of patients received the advised three follow-up visits in the first year after surgery. More follow-up visits were observed in younger patients and those diagnosed with a high-grade tumor, suggesting that doctors incorporate the patient’s risk of recurrence in their decision on follow-up frequency. The results of this study can help to improve follow-up practices, taking the risk of disease recurrence into account while avoiding unnecessary costs and tests.

**Abstract:**

Introduction: Follow-up (FU) in soft-tissue sarcoma (STS) patients is designed for early detection of disease recurrence. Current guidelines are not evidenced-based and not tailored to patient or tumor characteristics, so they remain debated, particularly given concerns about cost, radiation frequency, and over-testing. This study assesses the extent to which STS patients received guideline-concordant FU and to characterize which type of patients received more or fewer visits than advised. Methods: All STS patients surgically treated at the Leiden University Medical Center between 2000–2020 were included. For each patient, along with individual characteristics, all radiological examinations from FU start up to 5 years were included and compared to guidelines. Recurrence was defined as local/regional recurrence or metastasis. Results: A total of 394 patients was included, of whom 250 patients had a high-grade tumor (63.5%). Only 24% of patients received the advised three FU visits in the first year. More FU visits were observed in younger patients and those diagnosed with a high-grade tumor. Among patients with a recurrence, 10% received fewer visits than advised, while 28% of patients without a recurrence received more visits than advised. Conclusions: A minority of STS patients received guideline-concordant FU visits, suggesting that clinicians seem to incorporate recurrence risk in decisions on FU frequency.

## 1. Introduction

Soft-tissue sarcomas (STSs) are rare tumors, accounting for <1% of all tumors [1]. STSs are extremely heterogenous tumors with over 100 different histological subtypes and can affect people of all ages at any anatomical site [2,3]. Surgical excision is the standard treatment with or without adjuvant radiotherapy for local disease control [4]. Relapse following primary treatment occurs frequently, with 40–50% of STS patients developing either local or distant disease recurrence [5]. However, these percentages substantially differ per tumor grade, size, and subtype [6] and are highest in the first years after treatment [5,7,8]. Therefore, routine follow-up is designed to detect disease recurrence as early as possible because early treatment improves prognosis [4]. 

Long-term follow-up strategies have hardly been investigated in STS patients, and current guidelines are mostly based on expert opinions rather than on high-quality evidence [4]. Thus, the frequency and timing of follow-up and appropriate screening modalities continue to be the topic of debate, particularly as they are not tailored to the patient or tumor characteristics. Additionally, there are concerns about cost, radiation frequency, and possible over-testing [4,9,10]. Current ESMO-EURACAN (European Society for Medical Oncology- European Reference Network for rare adult solid cancers) and NCCN (National Comprehensive Cancer Network) consensus guidelines recommend follow-up visits every 3 to 6 months in the first three years, then twice a year up to the fifth year and annually thereafter, with NCCN guidelines distinguishing between low- and high-grade STS (Table 1) [4,11]. Survey data among musculoskeletal oncologists in the UK and orthopedic surgeons in the USA suggest high variation in current clinical practice regarding the number of follow-up visits [12,13]. However, it is unknown to what extent STS patients receive follow-up visits as recommended in guidelines and what type of patients receive more or fewer visits than advised, which may provide directions to improve care and limit overuse. 

Therefore, this study aimed to assess the extent to which STS patients after curative surgical treatment received guideline-concordant follow-up visits and to characterize which type of patients received more or fewer visits than advised. We distinguished between patients with and without an recurrence to identify possibilities for improvement of care, as fewer visits than advised among patients with a recurrence could point to possibilities to improve decisions on follow-up frequency for specific patient groups, and more visits than advised among patients without a recurrence could point to possible overuse.

## 2. Methods

### 2.1. Study Design and Patient Population 

In this retrospective cohort study, all STS patients surgically treated at the Leiden University Medical Center (LUMC) between 1 January 2000 and 1 January 2020 were included. The LUMC is one of the largest tertiary referral center for patients with sarcoma in the Netherlands. Patients with a sarcoma in the abdomen or thorax or on the face were excluded because a different follow-up schedule applies to these patients. The study protocol was presented to the Medical Ethics Committee of the Leiden University Medical Center, which waived the need for ethical approval under the Dutch law (W22.013). 

### 2.2. Definitions 

Start of follow-up was defined as the last day of treatment when curative status was achieved, which was either the last day of post-operative radiotherapy (RT) or date of surgery, with all patients followed up for 5 years thereafter. This time period has been indicated because it is during the first 2–3 years that recurrences are most likely to occur [4]. If patients were treated surgically more than once, the last operation before reaching complete remission was used. The local guideline for follow-up was similar to ESMO and NCCN guidelines, namely three visits per year (every 4 months) in the first three years and two visits per year (every 6 months) in the fourth and fifth year (Table 1). A follow-up visit was defined as having a radiological examination (i.e., either a PET-CT, MRI, or chest X-ray) so that all radiological examinations in the follow-up for each patient were included. The radiological examinations that occurred within two weeks of surgery were considered postoperative imaging rather than a follow-up moment and were therefore excluded from the dataset. As more than one radiological examination can be performed during one follow-up moment, the dates can differ for each radiological examination, as it is logistically challenging to schedule these on one day. Therefore, if the difference between two radiological examinations was less than 21 days, it was considered one follow-up moment.

A recurrence was defined as a local recurrence (LR), regional recurrence, or distant metastasis (DM). The time to recurrence was defined as the time from follow-up start to the first recurrence. The time at risk was defined as the time from follow-up start until (1) the first recurrence or (2) death or (3) 1 January 2020 or (4) the end of the 5-year follow-up period. Every death (not only as a direct result of disease) was included.

### 2.3. Patient, Tumor and Treatment Characteristics

Routinely available patient, tumor, and treatment characteristics from our local cancer registry were used: age, sex, histological subtype, tumor grade, and tumor size. In addition, we retrospectively collected surgical margins (R0/R1–2) and treatment with (neo)adjuvant radiotherapy (RT) (yes/no) from the medical records using a pre-specified data form. 

### 2.4. Statistical Analysis

Baseline characteristics were described using frequencies and percentages for categorical variables and mean (standard deviation) or median (interquartile range) for continuous variables, depending on the distribution. To indicate trends in follow-up over time, follow-up patterns were described for patients operated in three periods (2000–2005, 2006–2010, and 2011–2015) by time at risk and the extent to which each follow-up visit took place in the time window as advised in the guideline. Not all patients diagnosed between 2016–2020 completed the five-year follow-up and were therefore not visualized. 

To characterize the type of patients receiving more or fewer follow-up visits, we compared all patients diagnosed from 2000 until 2020 receiving more visits with those receiving fewer visits than advised in the guideline on the abovementioned patient (age, sex), tumor (grade and size), and treatment characteristics (surgical margins, RT). This was carried out by year of follow-up using an independent sample *t*-test in case of continuous variables and a chi-square test (or Fisher’s exact test) in case of categorical variables. These analyses were repeated separately for patients with and without a recurrence.

All analyses were conducted using R software, version 4.1.3 (R Foundation for Statistical Computing, Vienna, Austria) [14]. A *p*-value < 0.05 was considered statistically significant in all analyses.

## 3. Results

### 3.1. Patient Characteristics, Follow-Up Visits, and Incidence of Recurrence 

In total, 394 STS patients were included (Table 2). The median age was 60 years (IQR 46–71), and 51% was male. This group included 38 different diagnoses, including myxofibrosarcoma, liposarcoma, and leiomyosarcoma (combined 49% of the total patient population). The majority of patients had either a sarcoma in the lower extremities (69%), upper extremity (16%), or in the pelvis (8%), with a median tumor size of 7 cm (5–13). Overall, 250 patients had a high-grade tumor (63.5%) and 138 patients a low-grade tumor (35%), and for the remaining 6 patients (1.5%), the grade status was unknown. In most patients (*n* = 252), the tumor was removed with free surgical margins (R0, 63%). Fifty patients (13%) had more than one surgery before reaching complete remission. About one-quarter of patients (*n* = 102) were treated with radiotherapy, of whom 33 (32%) were treated pre-operatively and 69 (68%) post-operatively.

Figure 1 shows the follow-up visits over the course of each patient’s time at risk (depicted on each row) for the three periods. The orange bars indicate the advised timing of follow-up based on the guidelines. It is shown that the frequency of imaging substantially increased over the years, with patients having surgery in 2011–2015 receiving more imaging, particularly in the first year after treatment. Most visits occurred outside the orange bars and thereby the time intervals advised in the guidelines. 

A recurrence (distant or local) occurred in 110 patients (28%), with a median time to recurrence of 344 days (IQR 148–612). Moreover, 54% of recurrences occurred in the first year and 88% up until two years after surgery. Ninety-five patients (24%) died within the FU period, with a median of 663 days (IQR 318–1148) after surgery. The median age of a patient with recurrent disease was 64 years (IQR 46–74), and 52% was male. Among patient with a recurrence, 92 (84%) had a high-grade tumor compared with 16 patients with a low-grade tumor (14%). For two recurrences (2%), the grade status was unknown.

### 3.2. Type of Patients Receiving More or Fewer Follow-Up Visits

In the first year, 394 patients were at risk, with only 93 (24%) patients receiving the required three follow-up visits, while 185 (47%) patients had fewer visits, and 116 (29%) had more visits than advised in guidelines (Table 3). Patients having fewer visits than advised were significantly older than patients receiving more visits than advised (median age 62 vs. 54, *p* < 0.05) and were less often diagnosed with a high-grade tumor (43% vs. 87%, *p* < 0.05) or treated with RT (14% vs. 38%, *p* < 0.05). Surgical margins (*p* = 0.084) and tumor size (*p* = 0.221) did not significantly differ between patients receiving more and fewer visits than advised. Over the years, patients receiving more visits than advised were more likely to be patients with high-grade tumors compared with those receiving fewer visits, whereas the difference in age was smaller than in the first year. Similar patterns for free surgical margins and RT were observed in subsequent follow-up years but with fewer remaining patients at risk. 

### 3.3. Type of Patients Receiving More or Fewer Follow-Up Visits in Relation to Recurrence

Among the 394 patients at risk, 62 patients (16%) experienced a recurrence in the first year after surgery (Table 4). Among the patients experiencing a recurrence, 28 (45%) received the number of follow-up visits advised in the guidelines, 28 (45%) patients received more, and 6 (10%) had fewer follow-up visits, which could point to possibilities for improving decision making on FU frequency for some patient groups. Patients with a recurrence receiving fewer visits than advised were older (72 vs. 55 *p* < 0.05) than patients receiving more visits than advised in the guidelines but did not differ in histological grade (*p* = 0.415) or any of the other characteristics in the first year of follow-up.

In contrast, among patients without a recurrence, only 20% received the number of advised follow-up visits; 86 (28%) received more visits than advised, which may suggest possible over-testing; and 161 (51%) received fewer visits than advised in the guidelines (Table 5). Patients without a recurrence receiving more visits than advised were significantly younger than patients receiving fewer visits (53 vs. 61 years, *p* < 0.05), more often had a high-grade tumor (86% vs. 35%, *p* < 0.05), were treated with RT (42% vs. 14%, *p* < 0.05), and had free surgical margins (76% vs. 57%, *p* < 0.05). Tumor size (*p* = 0.37) did not significantly differ between patients receiving more and fewer follow-up visits than advised.

## 4. Discussion

The present study has shown that the frequency of imaging substantially increased over time, particularly in the first year after surgery, and that only one-quarter of patients received the advised three FU visits, with similar patterns in subsequent FU years. Patients receiving fewer visits than advised in the guidelines in the first year were significantly older and less often diagnosed with a high-grade tumor or treated with RT than those receiving more visits than advised. Among patients with a recurrence, 10% received fewer visits than advised and were older but did not differ on other tumor or treatment characteristics. About one-quarter of patients without a recurrence received more visits than advised and generally were younger, more often diagnosed with a high-grade tumor, and treated with RT or with free surgical margins. These observations may point to physicians incorporating recurrence risk in decisions on FU frequency but could also indicate areas for improvement.

There may be several explanations why physicians deviate from guideline recommendation: (1) present guidelines are not tailored to patient, tumor, or treatment characteristics; (2) there may be fear and uncertainty from patients’ or physicians’ perspectives; and (3) the effectiveness of follow-up to influence overall survival is uncertain. As a first explanation, current ESMO and NCCN guidelines do not distinguish between tumor -and treatment-related risk of disease recurrence, even though the risk for local or metastatic disease recurrence is known to differ significantly between STS patients due to tumor and treatment factors [4,11,15,16,17,18]. Our results also reflect this, showing that among patients experiencing a recurrence, 84% had a high-grade tumor. Not including known risk factors for a recurrence in follow-up strategies in current guidelines may be why physicians do not follow guidelines. This is consistent with survey results from Gerrand (2007) and Greenberg (2016), showing that physicians determined follow-up strategies based on perceived recurrence risk rather than adhering to guidelines, mostly because guidelines were not evidence-based [12,13]. Overall, the wide variation in follow-up visits observed in this study points to follow-up visits occurring almost at random, which should create awareness among physicians. Furthermore, it highlights the need for high-quality evidence supporting the current FU frequency in guidelines that is tailored to recurrence risk by including known risk factors to create risk classes, which may improve attitudes and beliefs concerning the guidelines.

As a second explanation, patients may express psychological stress and fear of recurrence particularly in the first year after treatment, and more frequent visits can reassure patients [19,20,21]. Additionally, when physicians experience uncertainty about the right course of action, they may also prefer to “err on the side of caution” [22] and give patients more frequent follow-up visits just to be safe. In our study, physicians’ fear or uncertainty may have played a role particularly in younger patients with higher life expectancies, as we found that patients who had more follow-up visits than advised were significantly younger than patients receiving fewer visits. Particularly, the 28% patients without a recurrence receiving more visits than advised might indicate possible overuse. Thus, anxiety or uncertainty, whether it comes from a patient or physician, may influence follow-up strategies, and this is an important factor of overuse [23,24]. Future initiatives seeking to reduce the number of follow-up visits should include interventions to address such uncertainty.

As a third explanation, despite the excessive increase in surveillance imaging in recent years, overall survival of STS patients has not improved, adding to existing doubts about the effectiveness of follow-up for early detection of recurrent disease [5,25,26], which may be why physicians do not follow guidelines but could point to underuse. For example, the 10% of patients with a recurrence receiving fewer follow-up visits than advised might indicate such underuse.

To the best of our knowledge, this is the first study that empirically quantified follow-up patterns of STS patients. Current clinical practice is an essential first step to guide subsequent initiatives to improve quality of care and eliminate overuse. One of the limitations is that its retrospective design as policies for diagnosis, treatment, and follow-up of STS patients may have changed over the past 20 years, which could have influenced our results. However, similar and seemingly random patterns were shown for different periods in which patients received treatment, indicating that the problem remained even for patients treated more recently. Another limitation is that results are based on a cohort of patients from a single center, which limits generalizability to other settings. On the other hand, our results seem consistent with survey results from other settings [12,13,27]. Therefore, issues regarding attitudes and beliefs about follow-up guidelines and uncertainty among healthcare professionals are likely similar elsewhere, which is why we think our study adds to the existing literature on this topic.

The results from our study may have implications for clinical practice in reducing overuse from a patient (i.e., radiation exposure and psychological burden) as well as healthcare perspective (i.e., resources and cost). The amount of follow-up visits is increasingly under debate not just for sarcoma patients but also for other oncological diagnoses [20,28,29]. The current “one-size-fits-all” approach may not be well suited for the different risks of recurrence in the heterogenous sarcoma population and thereby result in overuse. Our study gives some suggestions for which patient groups may be targeted to further explore and reduce overuse. Furthermore, the use of prediction tools such as Sarculator and Personalised Sarcoma Care (PERSARC) make more individualized prediction of LR/DM-risk possible [15,17,18,30,31,32]. This may facilitate clinicians in developing risk-based follow-up schedules, which can result in less-frequent follow-up visits for low-risk patients.

## 5. Conclusions

A minority of soft-tissue sarcoma patients received the advised three follow-up visits in the first year after surgery, and clinicians seemed to incorporate recurrence risk in their decisions on follow-up frequency rather than the “one-size-fits-all” approach given in guidelines. In addition, a significant proportion of patients without a recurrence received more follow-up visits than advised. Clinicians may therefore need support in more accurately estimating recurrence risk, e.g., by using prediction models to facilitate risk-based follow-up schedules, which can reduce over-testing of patients and consequently the burden on healthcare and costs.

## Figures and Tables

**Figure 1 cancers-15-04617-f001:**
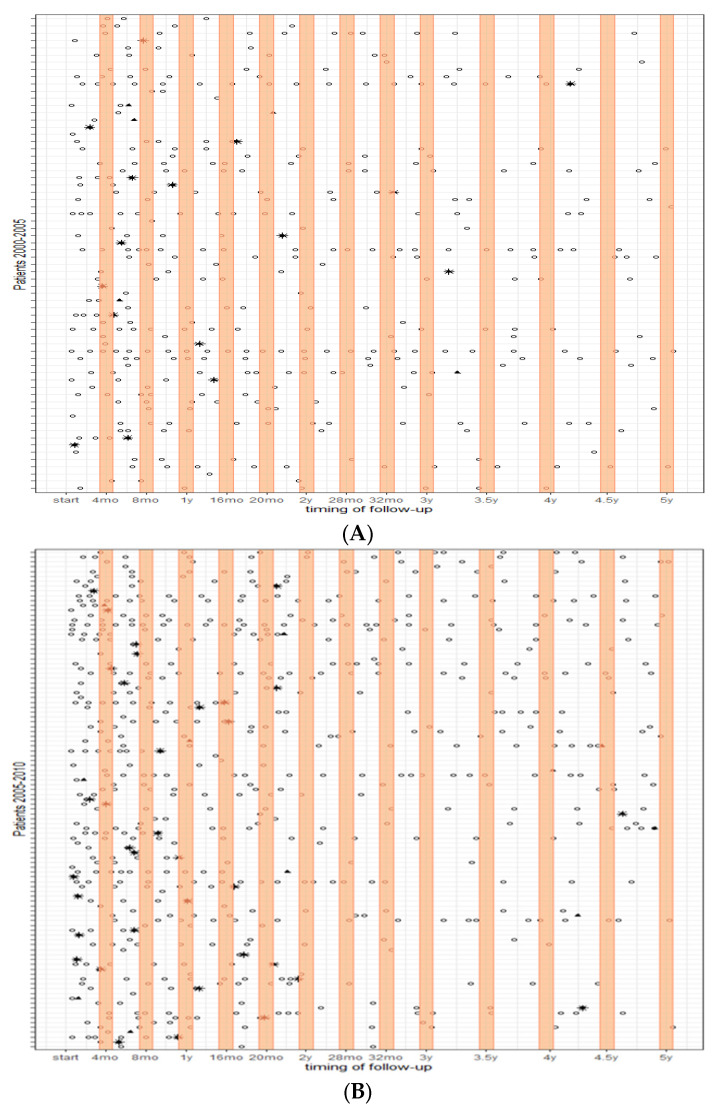
Follow-up patterns over 5 years; (**A**) (top) patients who had surgery between 2000–2005, (**B**) patients who had surgery between 2005–2010, and (**C**) (bottom) patients who had surgery between 2010–2015. Each row represents each patient’s time at risk; the orange bars indicate the advised timing of follow-up based on the guidelines. ○ = surveillance imaging; * = recurrence; ⯅ = maximum time at risk.

**Table 1 cancers-15-04617-t001:** Frequency of follow-up visits per guideline.

Follow-Up	Frequency of Visits in Years 1–3	Frequency of Visits in Years 4–5
ESMO	Every 3–4 months	Every 6 months
NCCN	Every 3–6 months	High grade—every 6 months Low grade—annually
Local guideline	Every 4 months	Every 6 months

**Table 2 cancers-15-04617-t002:** Patient characteristics and clinical-, pathological-, and treatment-related parameters (Leiden University Medical Center, N = 394).

Variables	Soft-Tissue Tumor Patients N (%)
Age at surgery (years) median ± IQR	60 (46–71)
Sex Male	200 (51)
Histology (most common) Myxofibrosarcoma Liposarcoma Leiomyosarcoma Myxoid liposarcoma Synovial sarcoma Spindle cell sarcoma Other	78 (20) 61 (16) 50 (13) 39 (10) 26 (6.6) 10 (3) 118 (30)
Tumor location Lower extremities Upper extremities Pelvis Other	272 (69) 64 (16) 31 (8) 28 (7)
Tumor size (cm) Median ± IQR	7 (5–13)
Grading High grade (II-III) Low grade (I) Unable to classify	250 (64) 138 (35) 6 (1.5)
Surgical margins R0 R1–2 Unknown	251 (63) 127 (32) 18 (5)
RTX Pre-operative Post-operative	N = 102 33 (32) 69 (67)

Abbreviations: IQR, interquartile range; RTX, radiotherapy.

**Table 3 cancers-15-04617-t003:** Observed n of follow-up compared to local guideline for soft-tissue sarcoma patients.

		Year 1	Year 2	Year 3	Year 4	Year 5
*n* at risk *		394	309	257	226	202
**Follow-up versus guidelines**						
	Fewer than guidelines	185 (47%)	161 (52%)	187 (73%)	191 (85%)	130 (64%)
	Following guidelines	93 (24%)	62 (20%)	38 (15%)	19 (8%)	44 (22%)
	More than guidelines	116 (29%)	86 (28%)	32 (13%)	16 (7%)	28 (14%)
**Median age (IQR)**						
	Fewer than guidelines	62 (50–73)	60 (47–70)	58 (44–69)	59 (46–70)	57 (44–69)
	Following guidelines	61 (43–72)	61 (48–72)	54 (40–68)	52 (41–65)	49 (39–63)
	More than guidelines	54 (42–67)	56 (43–68)	57 (47–64)	55 (43–63)	57 (51–64)
**Male gender in %**						
	Fewer than guidelines	49%	50%	49%	48%	53%
	Following guidelines	41%	44%	52%	66%	54%
	More than guidelines	61%	56%	63%	50%	55%
**High grade of tumor in %**						
	Fewer than guidelines	79 (43%)	105 (48%)	105 (48%)	64 (42%)	61 (42%)
	Following guidelines	70 (75%)	36 (72%)	17 (74%)	28 (60%)	25 (68%)
	More than guidelines	101 (87%)	31 (80%)	12 (75%)	21 (75%)	17 (85%)
**Tumor size (IQR)**						
	Fewer than guidelines	8 (5–13)	7 (4–13)	7 (4–13)	8 (5–13)	8 (4–13)
	Following guidelines	7 (4–12)	6 (5–11)	6 (5–10)	6 (4–10)	6 (5–12)
	More than guidelines	8 (5–14)	8 (5–12)	11 (5–17)	7 (5–15)	10 (3–17)
**Free surgical margins in %**						
	Fewer than guidelines	105 (61%)	134 (61%)	136 (62%)	91 (60%)	83 (64%)
	Following guidelines	60 (70%)	34 (68%)	20 (87%)	34 (72%)	32 (84%)
	More than guidelines	86 (71%)	30 (77%)	14 (88%)	22 (79%)	15 (75%)
**Radiotherapy in %**						
	Fewer than guidelines	27 (14%)	37 (17%)	48 (22%)	30 (20%)	33 (23%)
	Following guidelines	31 (33%)	22 (44%)	10 (44%)	21 (45%)	17 (46%)
	More than guidelines	44 (38%)	18 (46%)	8 (50%)	11 (40%)	12 (60%)

* deceased patients and patients with time at risk > 5 years are excluded for each year. Numbers in red indicate a significant difference between patients having more or fewer visits than advised.

**Table 4 cancers-15-04617-t004:** Type of STS patients with a recurrence receiving more or fewer visits than advised in guidelines.

		Year 1	Year 2	Year 3	Year 4	Year 5
***n* at risk ***		394	309	257	226	202
**Total recurrences ****		62	30	8	4	6
	Fewer than guidelines	6 (10%)	12 (40%)	5 (63%)	2 (50%)	4 (66%)
	Within guidelines	28 (45%)	12 (40%)	3 (37%)	2 (50%)	1 (17%)
	More than guidelines	28 (45%)	6 (20%)	0 (0%)	0 (0%)	1 (17%)
**Median age (IQR)**						
	Fewer than guidelines	72 (65–79)	73 (68–79)	69 (63–76)	71 (68–75)	56 (35–73)
	Within guidelines	60 (43–68)	68 (55–77)	64 (51–68)	63 (54–71)	64
	More than guidelines	55 (42–70)	64 (59–73)	-	-	23
**Male gender in %**						
	Fewer than guidelines	4 (67%)	4 (33%)	2 (40%)	-	3 (75%)
	Within guidelines	10 (36%)	3 (25%)	2 (67%)	2 (100%)	1 (100%)
	More than guidelines	21 (75%)	4 (67%)	-	-	1 (100%)
**High grade of tumor in %**						
	Fewer than guidelines	5 (83%)	10 (83%)	4 (80%)	-	2 (50%)
	Within guidelines	27 (96%)	7 (53%)	3 (100%)	1 (50%)	-
	More than guidelines	26 (93%)	6 (100%)	-	-	1 (100%)
**Tumor size (IQR)**						
	Fewer than guidelines	9 (7–11)	9 (6–12)	5 (5–7)	7 (4–9)	10 (9–12)
	Within guidelines	9 (5–15)	6 (5–8)	9 (7–10)	5 (5–6)	6
	More than guidelines	10 (6–15)	11 (7–14)	-	-	18
**Free surgical margins in %**						
	Fewer than guidelines	5 (83%)	6 (50%)	3 (60%)	1 (50%)	2 (50%)
	Within guidelines	15 (54%)	7 (58%)	2 (67%)	2 (100%)	-
	More than guidelines	17 (61%)	4 (67%)	-	-	1 (100%)
**Radiotherapy in %**						
	Fewer than guidelines	2 (33%)	3 (25%)	2 (40%)	-	-
	Within guidelines	13 (46%)	5 (42%)	2 (67%)	-	-
	More than guidelines	8 (29%)	3 (50%)	-	-	-

* deceased patients and patients with time at risk > 5 years are excluded for each year; ** either local, regional, or distant recurrence. Numbers in red indicate a significant difference between patients having more or fewer visits than advised.

**Table 5 cancers-15-04617-t005:** Type of STS patients without a recurrence receiving more or fewer visits than advised in guidelines.

		Year 1	Year 2	Year 3	Year 4	Year 5
***n* at risk ***		394	309	258	226	202
**No recurrences ****		336	258	226	202	168
	Fewer than guidelines	161 (51%)	182 (73%)	191 (85%)	130 (64%)	118 (70%)
	Within guidelines	62 (20%)	38 (15%)	19 (8%)	44 (22%)	35 (21%)
	More than guidelines	86 (28%)	32 (13%)	16 (7%)	28 (14%)	15 (9%)
**Median age (IQR)**						
	Fewer than guidelines	61 (49–72)	58 (44–69)	57 (44–68)	57 (45–70)	57 (44–67)
	Within guidelines	64 (44–72)	59 (46–69)	52 (38–68)	52 (40–65)	48 (39–62)
	More than guidelines	53 (39–64)	53 (44–68)	57 (47–64)	55 (43–63)	55 (51–62)
**Male gender in %**						
	Fewer than guidelines	78 (48%)	93 (50%)	97 (51%)	65 (50%)	60 (51%)
	Within guidelines	27 (44%)	19 (50%)	10 (53%)	28 (64%)	19 (54%)
	More than guidelines	49 (57%)	17 (53%)	10 (63%)	14 (50%)	8 (53%)
**High grade of tumor in %**						
	Fewer than guidelines	57 (35%)	81 (43%)	88 (46%)	55 (42%)	44 (37%)
	Within guidelines	41 (66%)	29 (29%)	13 (68%)	27 (61%)	25 (71%)
	More than guidelines	74 (86%)	24 (75%)	12 (75%)	21 (75%)	12 (80%)
**Tumor size (IQR)**						
	Fewer than guidelines	8 (5–14)	7 (5–13)	7 (4–13)	8 (5–13)	8 (5–14)
	Within guidelines	6 (4–9)	5 (4–12)	6 (5–10)	6 (4–13)	6 (5–13)
	More than guidelines	7 (4–13)	8 (4–12)	11 (5–17)	7 (5–15)	9 (3–17)
Free surgical margins in %						
	Fewer than guidelines	91 (57%)	117 (63%)	116 (61%)	76 (59%)	68 (58%)
	Within guidelines	42 (68%)	27 (71%)	17 (90%)	32 (73%)	30 (86%)
	More than guidelines	65 (76%)	26 (81%)	14 (88%)	22 (79%)	11 (73%)
**Radiotherapy in %**						
	Fewer than guidelines	23 (14%)	34 (18%)	46 (24%)	30 (23%)	23 (20%)
	Within guidelines	18 (29%)	17 (45%)	8 (42%)	21 (48%)	17 (49%)
	More than guidelines	36 (42%)	15 (47%)	8 (50%)	11 (39%)	10 (67%)

* deceased patients and patients with time at risk > 5 years are excluded for each year; ** either local, regional, or distant recurrence. Numbers in red indicate a significant difference between patients having more or fewer visits than advised.

## Data Availability

The data that support the findings of this study are available on request from the corresponding author upon reasonable request.

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
