# Peer review of "Follow-Up after Curative Surgical Treatment of Soft-Tissue Sarcoma for Early Detection of Recurrence: Which Patients Have More or Fewer Visits than Advised in Guidelines?"

_cancers, 2023, doi:10.3390/cancers15184617_

Round 1

Reviewer 1 Report

The Authors aimed to assess the extent to which STS patients received  guideline-concordant FU and to characterize which type of patients received more or less visits than advised. 

The topic is very interesting and the paper well organized.

Was follow up only up to 5 years?

which exams at each follow up?

were recurrences/metastasis detected earlier in the more strict follow up patients?Was prognosis different?

I would add which exams are recommended at each FU in table 1.

Fig 1 is confusing. Please detail further in the legend.

On the basis of these results, it would be an added value to see a proposed risk class scheduled follow up.

Author Response

We thank the editor and reviewers for their careful reading of the manuscript and constructive comments. We think these have considerably improved the manuscript. Below, we provide detailed responses to the comments. The line numbers refer to the revised document with track changes.

Yours sincerely,

Anouk Kruiswijk

Reviewer #1
The Authors aimed to assess the extent to which STS patients received  guideline-concordant FU and to characterize which type of patients received more or less visits than advised.
The topic is very interesting and the paper well organized.

  1. Was follow up only up to 5 years?

Response: We chose this duration of follow-up since it is during the first 2-3 years that recurrence are most likely to occur. We have added the rationale for choosing 5 years of follow-up in the methods section (page 2 lines 89-90).

  1. Which exams at each follow up?

Response: A follow-up visit was defined as having a radiological examination, either a PET-CT, MRI or chest X-ray. In our hospital imaging is always accompanied with physical examination. We clarified this in the method section, p3, line 94.

  1. Were recurrences/metastasis detected earlier in the more strict follow up patients? Was prognosis different?

Response: We agree with the reviewer that the impact of more follow-up visits on prognosis is an important issue, but the number of patients are too small to analyse this properly while also adjusting for relevant confounders such as tumor grade. As a first step, this paper therefore describes the extent to which patients receive guideline-concordant follow-up visits and tries to understand which type of patients received more or less follow-up than advised in the guidelines. The results of this study thereby might give some directions which patient groups may be targeted to further explore reasons for possible over- and underuse. It seems likely that there is an underlying reason why some patients receive more visits, i.e. known risk factors related to recurrence risk and that therefore time to recurrence might be shorter, rather than the earlier detection being the result of more strict follow-up. With our data, we were not able to distinguish between these two explanations, which would require a different type of study.

  1. I would add which exams are recommended at each FU in table 1.

Response: We appreciate the comment, however, our evaluation did not focus on the specific examination occurring at each follow-up point. Rather, we investigated the occurrence of any follow-up moment, which was identified by any radiological examination.

  1. Fig 1 is confusing. Please detail further in the legend.

Response: Figure 1 shows the follow-up visits over the course of each patient’s time at risk (depicted on each row) for the three specific time windows (2000-2005, 2005-2010, 2010-2015). The orange bars indicate the advised timing of follow-up based on the guidelines. We have added this explanation on page 7 lines 160-170. We agree the figure gives a messy picture, but that in fact reflects the pattern in patients receiving guideline-concordant follow-up.

  1. On the basis of these results, it would be an added value to see a proposed risk class scheduled follow up.

Response: We agree with the reviewer that the current ‘one-size-fits all’ approach may not be well suited for the different risks of recurrence in the heterogenous sarcoma population, and among the reasons why physicians do not follow the guidelines as described in the discussion. We have added more specifically that our results indicate the need for stratification based on known risk factors for recurrence (p10, lines 242-243).

Reviewer 2 Report

This is a very clear question with a clear answer

Very well written paper and very clearly written and analyzed with interesting results and conclusions

This is more of a descriptive paper than anything else with findings that are not so suprising in the sense that we image younger patients more and they more often have high grade tumours

The authors do mention that there is no data on improving OS with increased imaging so I would like to hear what they think guidelines should state or how they should be adapted 

Based on their findings what are their future perspectives on this work - do they want to establish a tailored surveillance guideline based on their findings etc. 

Author Response

We thank the editor and reviewers for their careful reading of the manuscript and constructive comments. We think these have considerably improved the manuscript. Below, we provide detailed responses to the comments. The line numbers refer to the revised document with track changes.

Yours sincerely,

Anouk Kruiswijk

  1. The authors do mention that there is no data on improving OS with increased imaging so I would like to hear what they think guidelines should state or how they should be adapted

Response: We thank the reviewer for reviewing our manuscript and the kind words. As discussed, we think that the currently used ‘one-size-fits-all’ approach in guidelines may not be suitable for the heterogenous sarcoma population. As a first step, guidelines could report follow-up frequency in risk classes based on known risk factors for recurrence, which we have now added more explicitly to the discussion (page 10 lines 242-243). In addition, clinicians could use prediction tools to make a more individualized prediction of LR/DM, to support them in creating risk-based follow-up schedules (page 10 lines 286-287).

  1. Based on their findings what are their future perspectives on this work - do they want to establish a tailored surveillance guideline based on their findings etc.

Response: Given the results of our study, it is clear that clinicians do not follow current guidelines and it would be of interest to understand their reasons in a more in-depth investigation. Since younger patients and patients with high-grade tumors have more frequent follow-up, it seems that clinicians already implicitly incorporate recurrence risk in their decision on follow-up frequency. As described above, we think that clinicians could be better supported in accurately estimating recurrence risk, e.g. by using prediction models to facilitate risk-based follow-up schedules.
